# Assessing the Knowledge of Analgesic Drugs Utilization during Pregnancy among Women in Saudi Arabia: A Cross-Sectional Study

**DOI:** 10.3390/ijerph18147440

**Published:** 2021-07-12

**Authors:** Maryam Jamal AlSaeed, Dalia Ahmed Elmaghraby

**Affiliations:** 1Department of Pharmacy Practice, College of Clinical Pharmacy, King Faisal University, Al Hofuf 31982, Saudi Arabia; delmaghraby@kfu.edu.sa; 2Titus Family Department of Clinical Pharmacy, School of Pharmacy, University of Southern California, 1985 Zonal Avenue, Los Angeles, CA 90089, USA

**Keywords:** analgesics, pregnancy, safety, effectiveness, knowledge, Saudi Arabia

## Abstract

Background: Pain is a common compensation mechanism in pregnant women that they may face during gestation due to physiological changes. Paracetamol and non-steroidal anti-inflammatory drugs are the most administered analgesic drugs worldwide. Therefore, safety and efficacy are important measures for the use of analgesics during pregnancy. Objective: Assess the knowledge of analgesic drug utilization among Saudi pregnant women. Method: We conducted a self-administered survey with an electronic questionnaire via Google Drive among a sample of 406 Saudi women. Results: About half of the respondents took analgesics during the first trimester, and 52.5% of women have used analgesics at least once without any medical advice during their gestation. Most participants agreed that paracetamol is the safest and effective analgesic drug during pregnancy, yet 61.8% of women are not aware that analgesics could be detrimental to the fetus if inappropriately administered in the third trimester. Conclusion: Participants have a good perception of the safest and most effective analgesic drug during pregnancy, but they have poor knowledge about analgesics’ side effects.

## 1. Introduction

Analgesic drugs are the most commonly administered drugs worldwide. A major reason accounting for their widespread use is their accessibility in the community setting; most analgesics are sold over the counter, and no prescription is required [1,2]. Examples of these accessible analgesics are paracetamol and non-steroidal anti-inflammatory drugs (NSAIDs), such as ibuprofen, diclofenac, naproxen, indomethacin, and aspirin. NSAIDs are frequently used by women of childbearing age, and pregnant and lactating women. As pain is expected due to physiological alteration during gestation, special concern should be had in terms of safety and effectiveness [3]. For selecting drugs based on their safety profile for pregnant women, healthcare decisions previously relied on pregnancy letter categorization (A, B, C, D, and X), yet this categorization does not represent the degrees of fetal risk and is not accurate, and confusing [4,5]. Therefore, the FDA has changed the labeling to be narrative and provide an informative summary of the designated drug, and the new labeling rule is called the “Pregnancy and Lactation Labeling Rule” (PLLR). PLLR became active on 30 June 2015 [4,5]. Here, the FDA released a notification concerning safety labeling for NSAIDs in which they can cause premature closure of the fetal ductus arteriosus, causing fetal renal dysfunction leading to oligohydramnios, in some cases, and neonatal renal impairment [4]. Taken together with these risks, the FDA suggests lower doses and duration of NSAIDs usage between 20 to 30 weeks of gestation and withdrawing their utilization from 30 weeks of gestation and later in pregnancy to prevent the possibility of developing oligohydramnios and premature closure of the fetal ductus arteriosus [4]. 

Concerning the risks related to trimesters, data suggest that NSAIDs’ administration before pregnancy or during the first trimester could increase the risk of abortion, unlike paracetamol [6,7]. Indeed, diclofenac and naproxen can cross the placenta at early gestation, and this increases the risk of abortion [7]. Although prenatal exposure to paracetamol in first and second trimesters has some reported adverse effects, such as developing cryptorchidism on newborn boys, it is still the safest analgesic for pregnant women and childbearing when given at the lowest effective dose and for the shortest duration possible [8,9,10]. Other studies showed that paracetamol as a single agent does not cause abortion in the first trimester, does not increase fetal risk in the first trimester, and is considered safe for use in pregnancy [7,11,12,13]. On the other hand, in the third trimester, NSAIDs have demonstrated their harmful effect on fetal development by inducing premature closure of the ductus arteriosus which leads to pulmonary arterial hypertension and respiratory problems [3,7]. Furthermore, oligohydramnios, neonatal anuria, and kidney dysfunction have been reported as risks induced by the use of NSAIDs in the third trimester of pregnancy, so NSAIDs should be avoided in the third trimester [3,7].

Several studies have been published recently about the knowledge, awareness, and perceptions of the general Saudi population towards NSAIDs, but specific subpopulations such as pregnant women and women of childbearing age were not addressed [1,14,15,16,17]. However, two Saudi studies investigated pregnant women’s knowledge about the use of NSAIDs and other medications such as paracetamol, antacids, antibiotics, and antihistamines [18,19]. Therefore, up until now, there has been no Saudi study that specifically examines pregnant or childbearing women’s knowledge of analgesic drug usage. Pregnant or childbearing women’s insight is pivotal because of the detrimental effects associated with NSAIDs’ administration in both the first and third trimesters, which may cause spontaneous abortions or fetal malformations, respectively. The purpose of the study is to assess the knowledge of Saudi women, who are pregnant or were previously pregnant, about analgesic drugs used during pregnancy.

## 2. Materials and Methods

### 2.1. Study Design, Sampling, and Data Collection

A cross-sectional survey was conducted in Saudi Arabia in March 2021. A self-administered questionnaire was distributed electronically in Arabic format only, since our target was Saudi women whose native language is Arabic. We used several types of social media, such as Twitter, Snapchat, WhatsApp, and Instagram, to distribute the questionnaire to reach our target population and sample size promptly and because of the public–related restrictions caused by the COVID-19 pandemic. 

### 2.2. Questionnaire Description

A self-administered questionnaire was provided via Google Drive. The questionnaire consisted of 28 questions divided into three parts. The first part included demographic information, such as age, educational status, medical field background, pregnancy status, number of pregnancies, and abortions. The second part assessed drug consumption patterns, the use of any over-the-counter (OTC) drugs, analgesic drugs- during the first trimester, and the source of medicines. In addition, respondents were asked to identify which analgesics they administered during the first trimester. Furthermore, we asked the participants about the administration of one of the drugs at least once without the doctor’s consent during pregnancy and the habits about taking these drugs before pregnancy for different reasons, such as dysmenorrhea, fever, and headache; and, if yes, what was the drug and its source.

The third part was focused on the assessment of knowledge related to the use, safety, and effectiveness of NSAIDs. Participants were required to identify which drug they considered NSAIDs, which one they frequently utilized during pregnancy, and which one they would think is better. In terms of pharmacological effects, the questions addressed whether using these NSAIDs will endanger the fetus in the third trimester or not; cause stomach ulcers, which NSAIDs have a detrimental effect on the stomach, and does paracetamol cause allergies? Here, we stated several brand names that were present in the Saudi market to ensure the proper understanding of the question. Some questions were asked about identifying which drugs may cause nausea, vomiting, and diarrhea, and the indications for using these drugs, such as headache, fever, rheumatic pain, and dysmenorrhea. Further topics included selecting the appropriate or the safest analgesic that can be used in the third trimester, and whether using NSAIDs and acetaminophen long-term during pregnancy would induce pulmonary atrial hypertension. Additionally, the participants were asked if they were educated about administering NSAIDs/analgesics- generally during pregnancy and what the source of their educational information was. The questionnaire used was adapted from previous studies and modified according to the particular Saudi environment [20,21]. The questionnaire was confirmed for its face and content validity by experts in the field of pharmacy practice and adjusted after a pilot study conducted on 10 participants.

### 2.3. Ethics

The ethical approval was obtained from the Research Ethics Committee (REC) at King Faisal University with the REC reference number (KFU-REC/2021-3-2). Confidentiality of participants’ information was assured. Informed consent for participation was given before administering the questionnaire. Additionally, a clear written statement stated that these data were intended for scientific research purposes.

### 2.4. Sample Size

We used the Raosoft^®^ sample size calculator, having a margin of error of 5% and a confidence level of 95%. The population was size-adjusted based on the subpopulation, women who experience pregnancy, that we are addressing. According to the report released by the General Authority for Statistics ©, Saudi females older than 18 years old number about 8,047,539 [22]. Here, the population size entered in Raosoft is 8,047,539, so the proposed sample is 385, and we recruited 406 women [23].

The sample size n and margin of error E are given by:
x = Z(c/100)2r(100 − r)
n = N x/((N-1)E2 + x)
E = Sqrt[(N − n)x/n(N − 1)]
where N is the population size, r is the fraction of responses., and Z(c/100) is the critical value for the confidence level c.

### 2.5. Statistical Analyses

Data collected were encoded and processed before analysis using MS Excel. Statistical analyses were conducted using JASP version 0.14 [24]. Frequencies and proportions expressed as percentages were generated for various qualitative variables including socio-demographic characteristics and aspects related to the use of analgesics before and during pregnancy. Appropriate graphs were also generated by GraphPad Prism^®^ version 9.00 for Mac OS (San Diego, CA, USA) [25].

To score the knowledge, the correct responses to the 19-item inventory of knowledge regarding analgesic use during pregnancy were added per respondent. For the correct responses, we coded as one value “scoring system”, while wrong answers were coded as zero. Hence, the total scores were then dichotomized as either low level of knowledge or high level of knowledge. A low level of knowledge corresponded to a score of lower than 10 points while a score of 10 to 19 was considered as a high level of knowledge. The proportions of respondents with high and low levels of knowledge were determined.

Logistic regression was carried out to determine the extent of the association between level of knowledge regarding analgesic use and various medico-socio-demographic variables. The variables age, educational attainment, number of pregnancies, and miscarriage experience were recategorized to prevent data sparsity among the initially identified categories. These variables together with the field of study and use of analgesics before pregnancy were individually subjected to the univariate logistic model. The odds ratios and their corresponding 95% confidence intervals were tabulated. A *p*-value of less than 0.250 was used as a cut-off to screen out independent variables for the succeeding multiple logistic regression analysis. Only the variables age, the field of study, and use of analgesic before pregnancy were included in the adjusted logistic regression model. 

## 3. Results

### 3.1. Participants Characteristics

A total of 406 Saudi women enrolled in our study. The inclusion and exclusion criteria in our study are as the following: we include (a) 18 years and older; (b) female; (c) Saudi nationality; (d) current or previous pregnancy history. We exclude (a) childbearing women who were never pregnant; (b) younger than 18 years; (c) non-Saudi women. Around 86.8% of respondents were between 26 to 46 years and older, including 26–35 years (31.8%), 36–45 years (28.6%), and 46 years or older (26.4%). In terms of the educational level of the participants, 64.5% of women held a bachelor’s degree, and 6.4% obtained higher education, master’s or/and Doctor of Philosophy. Although the majority of women are literate, 86% of them have no medical background. Pregnancy status revealed that 11.6% of women were pregnant when they responded to the questionnaire. Because we enrolled only women who have pregnancy experience, the pregnancy frequency of the participants was as follows: once (16.5%), twice (12.8%), three times (17.2%), and four times or more (53.4%). 52.7% have never encountered miscarriage, whereas 25.1% have experienced abortions at least once. Eight women only reported that their miscarriage was related to drug administration. The socio-demographic profile of the respondents is shown in Table 1.

### 3.2. Women’s Attitude towards Consuming Analgesic Drugs before Gestation

We assessed the women’s behaviors about consuming analgesic drugs in certain situations, such as influenzas and dysmenorrhea, before pregnancy. Thus, we found 329 out of 406 (81%) women claimed that they used analgesics before gestation, and the majority of women either self-medicated or relied on a physician’s prescription. Interestingly, for the source of prescription, 148 women self-medicated and 147 women used the analgesic per a physician’s order, which comprises 44.98% and 44.68%, respectively.

### 3.3. Drug Utilization Patterns before and during Gestation

Approximately half of the women (n = 206) declared that they took analgesics at the beginning of gestation, which is the first trimester. Out of 206, 173 of the women admitted to using paracetamol at the beginning of pregnancy (83.9%). The women who responded that they took analgesics in the early pregnancy and revealed that they used the medication as per the doctor’s prescription account for 65.5% (n = 135). Interestingly, 60% (n = 124) of the participants admitted that they have taken analgesic drugs at least once during pregnancy without referring to a medical doctor. The findings of analgesics utilization before and during pregnancy are shown in (Table 2)

### 3.4. Women’s Enlightenment and Perception about Analgesics

In investigating the women’s knowledge about analgesic paracetamol and NSAIDs, we asked the participants whether analgesic drugs, generally, have anti-inflammatory properties or not; 41.6% (n = 169) reported as no, which means analgesic drugs have no anti-inflammatory effects. Then, we asked them to categorize the drugs by writing both generic and brand names. Participants identified analgesics as NSAIDs as following: ibuprofen (n = 142; 35%), diclofenac (n = 132; 32.5%), naproxen (n = 94; 23.2 %), paracetamol (n = 78; 19.2 %), and aspirin (n = 76; 18.7%). Furthermore, 194 women admitted that paracetamol is not an NSAID, (Figure 1). Women assume that paracetamol is the safest and most effective analgesic drug that would provide pain relief during gestation (n = 330; 81.3%) and (n = 316; 77.8%) (Table 3). 

In terms of risk during the third trimester, only 28.3% (n = 115) of women believed that taking analgesics during the last three months of pregnancy has detrimental effects. Regarding uncertainty, 34.5% of women reported that long-term administration of NSAIDs could induce unwanted effects to the fetus, such as pulmonary arterial hypertension, whereas the remaining responses were between do not know, maybe, or no, which are 31.8%, 30%, and 3.7%, respectively (Figure 2). For the analgesic that is safe to use in the third trimester, more than half of the participants chose paracetamol (n = 238; 58.6%).

From a pharmacological perspective, 40.1% (n = 163) of the participants stated that they do not know that long-term usage of NSAIDs could cause stomach ulcers (Figure 3). Interestingly, 59.5% (n = 241) do not know which of the drugs would cause stomach ulcers. About half of women claimed that paracetamol has nothing to do with causing allergies (n = 214; 52.7%). 76.4% of women have no idea about which analgesic drugs can cause nausea, vomiting, and diarrhea. Furthermore, we assessed the cases or the indications that the participants used these drugs to manage. Responses regarding the indications for using analgesics include headache (80.8%), fever (85%), dysmenorrhea (57.4%), and rheumatic pain (47.8%). The participants were examined if they were educated and aware of analgesic drugs’ side effects and risks if they are utilized during pregnancy, and from where they obtained the information. Our findings regarding this showed that 53.7% were not aware of analgesic drugs’ side effects and risks during gestation. Among participants, those who aware and educated are 46.3% (n = 188), and 48.9% (n = 92 out of 188) obtained the information from doctors. 

As aforementioned, the correct responses to the 19-item inventory of knowledge regarding analgesic use during pregnancy were added per respondent. The total scores were then dichotomized as either low level of knowledge or high level of knowledge. A low level of knowledge corresponded to a score of lower than 10 points while a score of 10 to 19 was considered as a high level of knowledge. Table 4 presents the distribution of respondents according to the level of knowledge regarding analgesic use during pregnancy. Roughly, 4 for every 10 respondents do possess a high level of knowledge.

The above associations with the level of knowledge regarding analgesic use during pregnancy have described in the table below. As per age, we found that women who were at least 36 years old were almost 87% more likely to have a higher level of knowledge compared to those who were 35 years or younger while holding constant their field of study and use of analgesic before pregnancy. As per the health profession or medical field, after adjusting the odds ratio, women were 6.8 times more likely to have a higher level of knowledge regarding analgesic use during pregnancy while holding their age and analgesic use before pregnancy constant. Interestingly, we found that women who have used analgesics before pregnancy were two times likely to have a higher level of knowledge regarding analgesic use during pregnancy while keeping their age and field of study constant. Table 5 presents the crude and adjusted odds ratio and the corresponding *p*-value of the various predictors of the level of knowledge of analgesic use during pregnancy.

## 4. Discussion

Studies assessing females’ knowledge about drug use during pregnancy are limited. To our knowledge, this is the first study that was conducted to evaluate the Saudi women’s knowledge about analgesic use in pregnant women. Pain is a common complaint during pregnancy, especially lower back pain and abdominal pain [26,27,28,29,30,31]. Pain can be relieved by analgesics or by nonpharmacological interventions [32,33,34]. Due to the side effects of analgesics on the fetus and pregnant women, the guidelines recommend the usage of pharmacological treatments for pain during pregnancy should be for the shortest duration using the lowest effective dose [2,3,4,5].

The majority of the study participants used paracetamol as an analgesic during pregnancy, which is the recommended analgesic during pregnancy by different guidelines. Unfortunately, 61.8% of the participants have poor awareness about the effects and side effects of analgesics, especially NSAIDs. According to the U.S. FDA and RCOG, paracetamol is the safest analgesic for pregnant women and children when given at the lowest effective dose and for the shortest duration possible. All NSAIDs if needed should be used under medical supervision, not OTC for the targeted subpopulation, pregnant women, with the lowest dose and shortest duration [4,35]. Taken together, comparing the findings of our study regarding the usage of analgesics in the first trimester, 50% of women used analgesics in the first trimester of their pregnancy. Lower analgesic use was found in a previous study conducted in Nigeria to explore the medication utilization among pregnant women at a secondary health institution that showed that 34.4% took paracetamol and 4.6% took NSAIDs [36].

Several studies conducted in Saudi Arabia reported an alarmingly high prevalence of self-medication with analgesics [37,38,39,40,41,42]. Our study found the prevalence of self-medication with analgesics during pregnancy to be 28%. Similar results were found in the study conducted in Ethiopia exploring the self-medicated and Safety Profile of Medicines Used among Pregnant Women in a Tertiary Teaching Hospital, which showed that 27.0% of pregnant women reported taking at least one type of conventional medicine for self-medication, mainly analgesics 92.3% [43]. A lower percentage was found in another study conducted in Indonesia where only 11.7% of women self-medicated at least once during pregnancy [44]. However, the percentage of self-medication in our study was lower than that found in Tanzania, 46.24%, and Pakistan, 37.9% [45,46]. Many respondents in our study use paracetamol as an OTC analgesic during pregnancy (83.9%) and before pregnancy (79%). The same findings were found in a previous study in assessing the use of OTC analgesics among the general population in Saudi Arabia, where 73.4% used paracetamol [17].

Pharmacists and healthcare professionals have an important role in educating pregnant women about the adverse effects of drugs on fetuses and pregnant women during pregnancy. A study carried out in Saudi Arabia showed that pregnant women received drug information more from drug pamphlets than healthcare professionals [18]. Similar results were reported in a study assessing the public attitude and perception about analgesics and their side effects among the general population in Rafha and Riyadh, Saudi Arabia where 24% of the participants took their information about analgesics from drug pamphlets [47].

More than half of the participants in our study did not know the adverse effects of analgesics during pregnancy. Pregnant women’s literacy about analgesic use during pregnancy is integral as several studies confirmed some adverse effects to the fetus related to analgesic use during pregnancy [7,8]. Interestingly, a survey about the antenatal educational program was conducted in Riyadh, Saudi Arabia that addressed new and expected mothers. The researchers reported that women have low scores for both age and educational level [48]. Furthermore, data suggest that 80% of miscarriages or maternal mortality, generally, are preventable [48]. Nonetheless, a Saudi survey was undertaken to assess Saudi women’s knowledge and perception about the miscarriage’s etiology; interestingly, nothing was addressing analgesics nor medications [49]. Therefore, we recommend, as part of preventable measures for maternal health, that the Saudi Public Health provides appropriate and sufficient courses and educational programs for Saudi women, for all childbearing women. The urgent need for these programs has to do with the fact that Saudi women’s fertility rate has reduced to be 2.2 children per woman per 2021 data, and educational programs might mitigate the cost of losing their pregnancy [50]. Furthermore, health care professionals should provide effective patient education to every woman.

Some strengths in our study include that participants are from different cities in Saudi Arabia and different age groups, which make our findings more generalizable. However, there are some limitations to this study. The use of a convenience sampling technique and also an online survey for data collection could miss some targeted populations who do not use the internet. Therefore, the use of an online survey and convenience sampling technique may affect the generalizability of our results. Emphasizing limitations, because we performed an electronic survey, we cannot measure the response rate here. Further, we might not reach the population that has impoverished knowledge, so there could be selection bias because some women do not have access to social media. We might also have recall bias here because most of the women were previously pregnant (88.4%).

## 5. Conclusions

Study participants have substandard knowledge about analgesics as drugs. However, they have a good perception of electing which is the safest and effective analgesic drug during pregnancy. Nonetheless, both public health agencies and pharmacists must work further to boost and ensure appropriate patients counseling. Additionally, they should report any unwanted adverse or side effects that occur during gestation under the surveillance system in the Saudi Food and Drug Administration and/or Saudi Public health agency that care also for non-communicable diseases.

## Figures and Tables

**Figure 1 ijerph-18-07440-f001:**
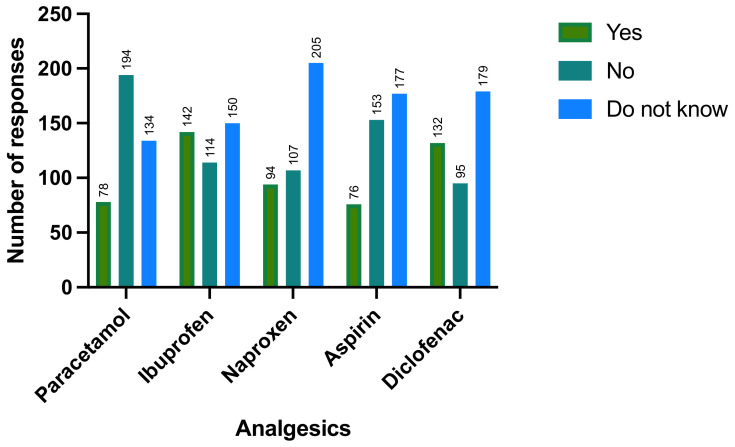
Are any of the following drugs considered non-steroidal anti-inflammatory drugs?

**Figure 2 ijerph-18-07440-f002:**
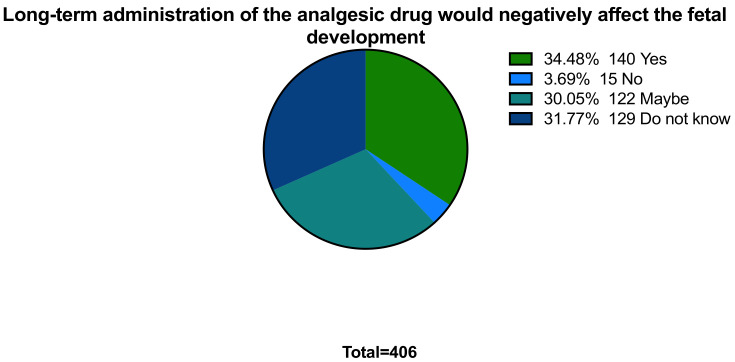
Long-term administration of the analgesic drug would negatively affect the fetal development.

**Figure 3 ijerph-18-07440-f003:**
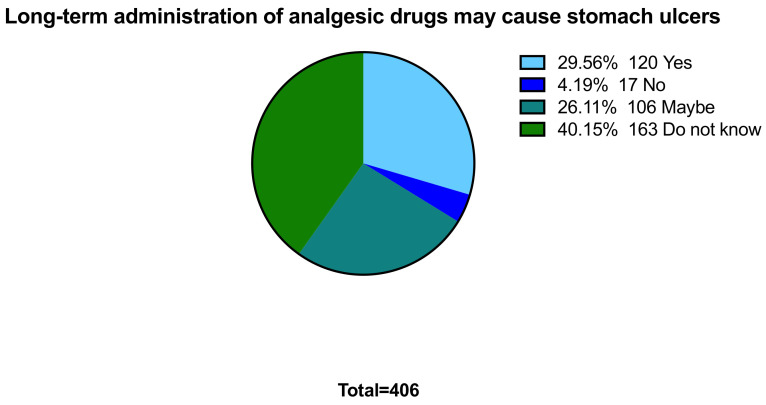
Long-term administration of analgesic drugs may cause stomach ulcers.

**Table 1 ijerph-18-07440-t001:** Socio-demographic profile of the respondents.

Socio-Demographic Characteristics	Count (n = 406)	Proportion, %
Age		
18 years and younger	4	0.985
19 to 25 years	50	12.315
26 to 35 years	129	31.773
36 to 45 years	116	28.571
46 years and older	107	26.355
Educational Attainment		
Elementary	7	1.724
High School	111	27.34
Bachelor	262	64.532
Master or Doctorate	26	6.404
Health professional		
Yes	57	14.039
No	349	85.961
Present Pregnancy status		
Yes	47	11.576
No	359	88.424
Number of Pregnancies		
Once	67	16.502
Twice	52	12.808
Thrice	70	17.241
Four or more times	217	53.448
Number of Miscarriages		
Never	214	52.709
Once	102	25.123
Twice	64	15.764
Three or more times	26	6.404
Drug-related Miscarriage (n = 288)		
Yes	8	2.778
No	234	81.25
Not sure	46	15.972

**Table 2 ijerph-18-07440-t002:** Use of analgesics before and during pregnancy.

Aspects Related to Use of Analgesics during Pregnancy	Count	Proportion, %
Use of analgesics during the first trimester (n = 406)		
Yes	206	50.739
No	200	49.261
Prescriber of the analgesic during pregnancy (n = 206)		
Physician	135	65.534
Pharmacist	9	4.369
Self-prescription	58	28.155
Family or friends	4	1.942
Specific analgesic is taken during pregnancy (n = 206)		
Paracetamol	173	83.98
Ibuprofen	1	0.485
Diclofenac	2	0.97
Naproxen	3	1.456
Aspirin	26	12.621
Others	1	0.485
Emergency use of analgesic without referral to a physician or pharmacist during pregnancy (n = 206)		
Yes	124	60.194
No	82	39.806
The usual use of analgesics before pregnancy (n = 406)		
Yes	329	81.034
No	77	18.966
Specific analgesic taken prior to pregnancy (n = 329)		
Paracetamol	260	79.027
Ibuprofen	24	7.295
Diclofenac	13	3.951
Naproxen	1	0.304
Aspirin	25	7.599
Others	6	1.824
Prescriber of the analgesic during pregnancy (n = 329)		
Physician	147	44.681
Pharmacist	23	6.991
Self-prescription	148	44.985
Family or friends	11	3.343
Been informed of the adverse effect of analgesics during pregnancy (n = 406)		
Yes	188	46.305
No	218	53.695
Source of information regarding adverse effects (n = 188)		
Physician	92	48.936
Pharmacist	15	7.979
Family or friend	31	16.489
Social media	32	17.021
Books	15	7.979
No response	3	1.596

**Table 3 ijerph-18-07440-t003:** Safety and effectiveness of analgesic during pregnancy.

Drugs	Safety (n/%)	Effectiveness (n/%)
Ibuprofen	12 (3)	27(6.7)
Naproxen	9 (2.2)	10 (2.5)
Paracetamol	330 (81.3)	316 (77.8)
Aspirin	49 (12.1)	40 (9.9)
Diclofenac	6 (1.5)	13 (3.2)

**Table 4 ijerph-18-07440-t004:** Distribution of Respondents according to the Level of Knowledge regarding Analgesics Use During Pregnancy.

Level of Knowledge	Count (n = 406)	Relative Frequency, %
Low (scores below 10 points)	251	61.823
High (scores of at least 10 points)	155	38.177

**Table 5 ijerph-18-07440-t005:** Crude and adjusted odds ratio of the association between level of knowledge regarding analgesic use during pregnancy and various medico-socio-demographic variables.

Predictors	Crude Odds Ratio	95% Confidence Interval	*p*-Value	Adjusted Odds Ratio ^a^	95% Confidence Interval	*p*-Value
Age						
35 years old and younger	1					
36 years old and older	1.182	0.931—2.095	0.107	1.875	1.194—2.945	0.006
Educational attainment						
Below College graduate	1					
College graduate and higher	0.949	0.712—1.730	0.645			
Field of Study						
Non-medical related	1	2.843—9.815	<0.001	6.802	3.527—13.120	<0.001
Medical-related	2.298					
Number of pregnancies						
Less than 3	1					
3 or more	1.01	0.626—1.508	0.898			
Miscarriage’s experience						
Without	1					
With	1.058	0.750—1.672	0.581			
Use of Analgesics before pregnancy						
Non-user	1					
User	1.408	1.139—3.454	0.016	2.019	1.116—3.652	0.02

Note: ^a^ Only those predictors with a *p*-value less than 0.25 were included in the adjusted model.

## Data Availability

The data presented in this study are available within the article.

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
