# Peer review of "Assessing the Knowledge of Analgesic Drugs Utilization during Pregnancy among Women in Saudi Arabia: A Cross-Sectional Study"

_ijerph, 2021, doi:10.3390/ijerph18147440_

Round 1
Reviewer 1 Report
Additional comments:
1. The methodology shows that the custom made questionnaire was used. Was the questionnaire validated before the survey? Have authors tested the validity and reliability of the scale?
2. The description of the statistical analysis used is very poor.
3. How was the size of the study group determined. Is the size of the study group (406 women) given in the methodology representative of the total female population in Saudi? Please provide evidence that the test group is representative.
4. What were the inclusion and exclusion criteria for patients from the study? 5. Part of the manuscript results should be re-written, for better understanding of the text, I propose to present the characteristics of the study group first, then to present the level of knowledge of the study group, and finally to present the logistic regression results.
6. Discussion section is very poor. It should be enriched with more references. The strengths and limitation of the conducted research lack in the discussion.
7. Conclusion section is impoverished, inconsistent with the purpose of the study.
Author Response
Please check the attachment
Sincerely,
Maryam Al Saeed

Reviewer 2 Report
1. What does Medical-related major mean in Table 1?2. Clarify the purpose of the study
3. What is the value of this study?
Author Response

(The authors gave the same response as above.)

Reviewer 3 Report
This study is to assess the knowledge of analgesic drug utilization among Saudi pregnant women using electronic questionnaire via Google drive. There are some major issues to be published.
- Please add information regarding sample size in this study. As this study is cross-sectional study and the appropriate sample size calculation should be conducted.
- In the discussion part, there is no limitation in this study, however, this self administered survey has some limitation, such as recall bias, etc. Please add bias issues, such as selection bias and the related limitation in this survey.
- The author needs to discuss about the next step for this study, such as how to link this study results to the public health action in Saudi. The current version of this study is presented the survey results, but not for public health massage, such as what kind of action is needed to improve the low score participants.
- Please rewrite more detail the statistical part. For example how to select variables for the logistic model? Please add 95%CI in the Table. Please add how to calculate score which you indicated in the statistical part.
- Please refine the Figures more appropriate, such as add Y axis definition in Figure 1.
- Please add some additional discussion, such as situation in Saudi for usage of analgesics ,etc. The current version is too few discussion points.
Author Response

(The authors gave the same response as above.)

Round 2
Reviewer 1 Report
I thank the authors of the paper for taking into account all my comments. The work has undergone significant innovation and can be published in its present form.
Author Response

(The authors gave the same response as above.)

Reviewer 2 Report
Please change the legend of Figure as follows;
Figure 1. Do any of the following drugs consider NSAIDs?
Figure 1. Long-term ~~~ fetal development.
Table 4. Distribution of ~~~ analgesics use during pregnancy
Author Response

(The authors gave the same response as above.)

Reviewer 3 Report
The quality of the revised version was much improved from the previous version.
Minor revision
Line 152 Please correct the terminology simple logistic model to the univariate logistic model.
Author Response

(The authors gave the same response as above.)
